# Hydration and Mechanical Properties of Low-Carbon Binders Using CFBC Ash

**DOI:** 10.3390/ma18122731

**Published:** 2025-06-10

**Authors:** Young-Cheol Choi

**Affiliations:** Department of Civil and Environmental Engineering, Gachon University, Seongnam 13120, Gyeonggi-do, Republic of Korea; zerofe@gachon.ac.kr; Tel.: +82-31-750-5721

**Keywords:** fluidized bed combustion ash, desulfurization gypsum, self-hardening property, hydration, compressive strength

## Abstract

Circulating fluidized bed combustion (CFBC) ash, a byproduct typically generated from coal-fired CFBC power plant boilers, contains high content of free lime and anhydrite. Due to its chemical composition, CFBC ash exhibits self-cementing properties; however, its performance is limited. One approach to enhancing the self-cementing properties of CFBC ash is through the incorporation of mineral admixtures such as gypsum. This study investigated the influence of desulfurization gypsum (DG) on the self-cementing behavior of CFBC ash. To this end, paste and mortar specimens were prepared and evaluated for their hydration and mechanical characteristics. The hydration behavior was analyzed using isothermal calorimetry, thermogravimetric analysis (TGA), setting time measurements, and X-ray diffraction (XRD) analysis. Mechanical properties were assessed by measuring the compressive strength at various curing ages. Additionally, changes in microstructure were examined by evaluating the pore size distribution through mercury intrusion porosimetry (MIP). The experimental results indicate that the appropriate incorporation of DG enhances the hydraulic reactivity of CFBC ash and significantly improves the compressive strength.

## 1. Introduction

Globally, fossil fuels continue to dominate as primary energy sources due to their current reserves, existing infrastructure, and relatively low fuel costs. However, traditional coal combustion technologies are known to emit sulfur dioxides (SO_x_) and nitrogen oxides (NO_x_), which contribute directly to air pollution [1]. In recent years, the adoption of circulating fluidized bed combustion (CFBC) technology has been gaining attention for its high combustion efficiency and its significant reduction in SO_x_ and NO_x_ emissions, thereby contributing to improved air quality [2,3]. With the increasing deployment of CFBC technology, the amount of CFBC ash generated as a byproduct is also growing substantially, raising concerns regarding its environmental impact and the need for effective utilization strategies [4]. Unlike conventional pulverized coal combustion fly ash (FA), CFBC ash is typically produced at a relatively low combustion temperature of around 900 °C. As a result, CFBC ash tends to exhibit an irregular particle morphology rather than spherical, and it generally contains a higher level of unburned carbon [5,6].

Furthermore, due to the addition of limestone during the desulfurization process, CFBC ash contains elevated levels of CaO and SO_3_, resulting in higher free CaO content compared to FA. This composition imparts excellent self-cementing properties to the CFBC ash, enabling it to function as a binder material with performance similar to that of cement [7,8,9]. Additionally, its chemical similarity to fly ash suggests potential pozzolanic reactivity, indicating its applicability as a supplementary cementitious material (SCM) [10,11]. However, the high free CaO content can lead to excessive heat evolution, and the presence of SO_3_ may induce undesirable expansion, both of which can adversely affect the performance of cement and concrete products [12].

Numerous researchers have investigated the hydration reactions of CFBC ash [13,14,15]. CFBC ash, which exhibits self-cementing properties, typically forms calcium silicate hydrate (C-S-H), ettringite, and Ca(OH)_2_ as its primary hydration products, which are similar to those observed in ordinary Portland cement (OPC) systems [6,14,16]. In CFBC ash with high SO_3_ content, the rapid formation of ettringite in the early stages of hydration has been reported to cause significant expansion. Sheng et al. [16] investigated the effects of the particle size on the self-cementitious properties of CFBC ash. According to their study, a lower 45 μm sieve residue of CFBC ash led to increased self-cementitious strength and contributed to the mitigation of volumetric instability. Sheng et al. [17] investigated the influence of lime content in CFBC ash on its self-cementing behavior and concluded that the formation of Ca(OH)_2_ through the slaking of free lime plays a dominant role in the development of self-cementing properties. Several studies have explored the use of CFBC ash in various construction materials, aiming to utilize it as a value-added component [18,19,20,21,22,23,24,25,26]. Due to its production characteristics, CFBC ash is expected to exhibit pozzolanic reactivity when blended with cementitious materials, which enhances its potential as a mineral admixture. The applicability of CFBC ash as an SCM has been validated by many researchers [27,28,29,30]. Shen et al. [31] further demonstrated that, due to the high SO_3_ content in CFBC ash, it could serve as a substitute for conventional gypsum in cement formulations, acting as a set-retarding agent in hydration processes.

In recent years, numerous studies have been conducted to significantly reduce environmental impacts by utilizing CFBC ash in the development of low-cement or zero-cement binders and related products, either by drastically reducing or entirely eliminating the use of OPC [32,33,34]. These studies primarily focus on blending CFBC ash with ground granulated blast furnace slag—a common precursor in alkali-activated cement systems—or activating CFBC ash directly using alkaline agents such as sodium carbonate or calcium hydroxide to formulate cementless binders. Moreover, owing to its inherent mineralogical composition and morphological advantages, CFBC ash has also been investigated for use in environmental applications such as heavy metal removal from contaminated media [35]. However, its practical application in the construction industry remains limited. This is largely due to issues such as excessive heat release during early hydration and expansion-induced cracking, which stem from the high content of free CaO and SO_3_ in CFBC ash [36]. Wang and Song [5] reported that the volumetric stability of CFBC-based cementitious systems can be maintained if the SO_3_ content is controlled below 3.5%. Gao et al. [37] conducted a study on SCMs containing CFBC ash in concrete. According to their research, f-CaO and SO_3_ in CFBC ash synergistically enhanced early hydration when combined with FA and ground granulated blast furnace slag (GGBS), with the optimal mix being 30% CFBC ash, 60% GGBS, and 10% FA. Rust et al. [38] conducted a study on the development of low-energy cement using CFBC ash and flue gas desulfurization (FGD) gypsum. They reported that FGD gypsum converted to hemihydrate effectively promoted early strength development in the CFBC ash-based binder. CFBC ash-based binders offer significant potential as low-carbon alternatives for sustainable construction by reducing the consumption of cement. However, to fully realize this potential, it is essential to enhance the self-cementing properties of CFBC ash.

In this study, the effect of DG on the self-cementing characteristics of CFBC ash was investigated. The hydration characteristics were evaluated through isothermal calorimetry, thermogravimetric analysis (TGA), setting time tests, and X-ray diffraction (XRD) analysis. Furthermore, mechanical performance was assessed by measuring the compressive strength at various curing ages. Microstructural evolution was also examined by analyzing the pore size distribution of specimens using mercury intrusion porosimetry (MIP) at different ages.

## 2. Experimental Details

### 2.1. Materials

In this study, CFBC ash was obtained from a CFBC boiler using bituminous coal as fuel, operated at a combustion temperature range of 800–950 °C. The ash was collected from the electrostatic precipitator after combustion. As shown in Figure 1, the CFBC ash exhibited a light brown color. The effects of DG on the hydration behavior of CFBC ash were investigated. The DG used in this study was obtained from Company H in South Korea.

Figure 2 presents the SEM images of CFBC ash and DG. While some spherical particles are observed in CFBC ash, the majority of the particles exhibit angular, fractured shapes. The DG particles exhibit irregular shapes with a wide range of sizes and relatively rough surface textures.

The chemical compositions of CFBC ash and DG were analyzed using X-ray fluorescence (XRF) spectroscopy, and the results are summarized in Table 1. Here, LOI stands for loss on ignition. The free CaO content was also determined using the ethylene glycol extraction method [39]. CFBC ash was found to primarily consist of CaO, SiO_2_, Al_2_O_3_, and Fe_2_O_3_, with SO_3_ content of 5.57%. The free CaO content was measured to be 11.1%. The free CaO and SO_3_ content in CFBC ash is attributed to the in-furnace desulfurization process, in which limestone is directly injected into the combustor, resulting in the formation of CaO and CaSO_4_. For DG, CaO and SO_3_ were the predominant components, accounting for 62.4% and 22.7%, respectively, and the free CaO content was significantly high at 44.5%. The specific gravities of CFBC ash and DG were 2.85 g/cm^3^ and 2.61 g/cm^3^, respectively.

Figure 3 shows the XRD patterns of CFBC ash and DG. Based on the XRD patterns in Figure 3, a quantitative phase analysis was conducted using the Rietveld refinement method. Corundum was used as the internal standard for the analysis. The phase composition of CFBC ash was determined to be 22.7% SiO_2_, 27.2% CaO, and 19.1.0% CaSO_4_. In the case of DG, the main phases were 41.3% CaO and 30.7% CaSO_4_. Additionally, Ca(OH)_2_ and CaCO_3_ were identified, which was attributed to partial hydration and carbonation due to exposure to atmospheric moisture.

Figure 4 presents the particle size distributions of CFBC ash and DG. The average particle sizes of CFBC ash and DG were measured to be 17.3 μm and 19.6 μm, respectively.

### 2.2. Mixture Proportions and Test Methods

In this study, paste and mortar specimens were prepared based on the mixture proportions shown in Table 2 to investigate CFBC ash-based binders as a substitute for cement. The primary variable was the replacement ratio of DG, aimed at evaluating its effects on the self-cementing behavior of CFBC ash. To examine the influence of the DG content, binder systems were prepared with DG replacing cement at 5%, 10%, and 15% by weight. To ensure the homogeneous mixing of CFBC ash and DG, the dry powders were thoroughly blended using a V-type mixer prior to specimen fabrication. For all specimens, the water-to-binder ratio was fixed at 0.5 by weight. ISO-standard sand was used for mortar specimens at a sand-to-binder ratio of 3:1 by weight.

The fresh mortar was prepared according to Table 2; then, it was placed into 40 mm × 40 mm × 160 mm prismatic molds and compacted using a vibration table to eliminate entrapped air; it was then finished to a smooth surface. After casting, the specimens were cured at 20 ± 1 °C and relative humidity greater than 90% for 24 h. Following demolding, the specimens were submerged in water for continued curing. For compressive strength testing, the specimens were cut in half to obtain 40 mm × 40 mm × 80 mm prisms, and a compressive load was applied uniformly to the surfaces. The final compressive strength values for each variable were determined by averaging six replicate measurements. The setting time (initial and final) was measured according to ISO 9597 [40].

The heat of hydration was measured under isothermal conditions using a TAM Air isothermal calorimeter (TA Instruments, New Castle, DE, USA). Approximately 4 g of paste, prepared according to the mixture proportions in Table 2, was sealed in an ampoule and placed in the calorimeter. The heat flow due to hydration was continuously recorded for 24 h at a constant temperature of 23 ± 0.1 °C. X-ray diffraction (XRD) patterns were obtained using a PANalytical X’Pert Pro MPD diffractometer (Malvern PANalytical, Almelo, The Netherlands) to analyze the crystalline phases. Measurements were conducted with Cu-Kα radiation over a 2θ range of 5° to 65°, using a step size of 0.02° and a scan rate of 0.5°/min. Thermogravimetric analysis (TGA) was performed using a Thermo plus EVO2 thermal analyzer (Rigaku, Tokyo, Japan) under a nitrogen atmosphere. Approximately 10 mg of a powder sample was heated from room temperature to 1000 °C at a heating rate of 10 °C/min. The mass loss of the sample was continuously recorded as a function of the temperature. The pore structure analysis of the paste specimens was conducted using a Micromeritics Autopore IV 9500 mercury intrusion porosimeter (Micromeritics Instrument Corporation, Norcross, GA, USA). Before testing, the samples were thoroughly dried and then mounted in the sample cell of the instrument. The pore size distribution was measured over a pressure range of 0.003 psia to 30,000 psia.

## 3. Results and Discussion

### 3.1. Hydration of CFBC Ash Containing DG

Figure 5 presents the normalized heat flow and cumulative heat release profiles of CFBC ash paste samples incorporating various amounts of DG. In typical OPC systems, a sharp exothermic peak—referred to as the initial hydrolysis peak—appears immediately upon contact with water. This is followed by a dormant period, after which the hydration of tricalcium silicate (C_3_S) begins in earnest, leading to the formation of C-S-H gel and Ca(OH)_2_ during the acceleration period. As the reaction rate gradually decreases, a distinct second peak (the main hydration peak) typically appears. However, here, CFBC ash exhibited different hydration behavior compared to OPC. As shown in Figure 5a, the CFBC ash system exhibited an initial peak approximately 5 min after water contact—similar in timing to OPC—but no distinct second peak was observed. This suggests a deviation from the typical C_3_S-driven hydration process found in OPC. As the DG content increased, the magnitude of the initial peak also slightly increased, accompanied by an overall increase in the peak area. This phenomenon was likely attributed to the presence of free CaO in the DG, which reacts exothermically with water.

Figure 5b illustrates the normalized cumulative heat released over time. The plain sample composed solely of CFBC ash showed a 24 h cumulative heat release rate of approximately 38.5 J/g. In contrast, samples containing DG exhibited higher cumulative heat release. Notably, the sample in which 20 wt% of CFBC ash was replaced with DG (CFDG20) exhibited a cumulative heat release rate of 45.6 J/g at 24 h—an 18.4% increase compared to the plain sample.

Figure 6 shows the XRD patterns of the plain sample cured for 7 and 28 days. As shown in Figure 6, the main hydration products identified in the CFBC ash system were portlandite and ettringite. In addition, unreacted components inherent in CFBC ash, such as CaO, quartz, anhydrite, and gypsum, were partially detected. In OPC systems, the ettringite formed during early hydration typically converts to monosulfate as the sulfate content diminishes and tricalcium aluminate (C_3_A) reacts further. However, in the CFBC ash system, monosulfate was not observed, likely due to the relatively high sulfate content and the low C_3_A concentration in the CFBC ash compared to OPC. Generally, the free CaO present in CFBC ash reacts with water to produce Ca(OH)_2_, which subsequently reacts with alumina or silica phases to form ettringite and C-S-H gel. These hydration reactions confer self-cementing properties to CFBC ash [7,8,9].

Figure 7 presents the XRD analysis results of CFBC ash pastes with varying DG content. The overall hydration products were similar to those observed in the plain sample (Figure 6). As shown in Figure 7a, at 7 days of curing, both the anhydrite peak near 25.5° and the portlandite peak around 34° increased with increasing DG content. This trend is attributed to the hydration of free CaO contained in DG. At 28 days, as illustrated in Figure 7b, the intensity of the ettringite peak increased compared to that at 7 days, while the intensities of the anhydrite and portlandite peaks decreased. These changes suggest that anhydrite may have undergone phase transformation through hemihydrate to dihydrate gypsum, which then reacted with portlandite, Al_2_O_3_, and water to form ettringite.

Figure 8 shows the TG curves of CFBC ash pastes with varying DG content at different curing ages. Weight losses were observed near 100 °C and 450 °C. The weight loss around 100 °C is generally attributed to the evaporation of free water and the decomposition of hydration products such as ettringite and C-S-H gel that had not reacted with the CFBC ash [41,42]. The weight loss near 450 °C corresponds to the thermal decomposition of Ca(OH)_2_, a key hydration product [41,42]. As the curing age increased, a greater degree of hydration occurred, leading to more significant weight loss near 100 °C. Additionally, as shown in Figure 8, the amount of Ca(OH)_2_ formed from the hydration of free CaO decreased over time, as indicated by the reduced weight loss near 450 °C. This suggests that Ca(OH)_2_ gradually transformed into ettringite and C-S-H gel over time.

Figure 9 presents the quantitative results regarding the Ca(OH)_2_ content derived from the thermogravimetric analysis shown in Figure 8. As illustrated in the graph, the Ca(OH)_2_ content increased with higher DG replacement levels, regardless of the curing age (excluding the 28-day value for the plain sample). At 7 days, the Ca(OH)_2_ content in the plain sample was 4.63%, while the sample with 20 wt% DG (CFDG20) exhibited Ca(OH)_2_ content that was approximately 2.73 times greater than that of the plain sample. With increasing curing ages, the Ca(OH)_2_ content generally decreased. In the plain sample, the Ca(OH)_2_ content decreased by 0.62% from day 7 to day 14, and it further decreased by 0.74% from day 14 to day 28. In contrast, samples containing DG exhibited a slightly different trend. Although the Ca(OH)_2_ content also decreased over time, the degree of reduction was not significantly affected by the DG replacement level. On average, the Ca(OH)_2_ content in DG-blended samples decreased by 0.74% from day 7 to day 14 and by 2.8% from day 14 to day 28.

### 3.2. Setting Time and Compressive Strength

Figure 10 illustrates the changes in the initial and final setting times of CFBC ash paste as a function of the DG content. Both the initial and final setting times increased with higher DG replacement levels. Specifically, as the DG content increased from 0% to 20%, the initial setting time increased from 48 min to 70 min, while the final setting time increased from 94 min to 117 min. This trend indicates that DG exhibits a retarding effect on the setting behavior of CFBC ash paste.

To evaluate the effects of the DG content on the mechanical performance of the CFBC ash mortar, compressive strength tests were conducted at curing ages of 3, 7, 14, and 28 days. The results are presented in Figure 11. DG was incorporated at varying levels of 0%, 5%, 10%, 15%, and 20% by weight. As shown in Figure 11, the development of the compressive strength was significantly influenced by the DG content. At early ages (3 and 7 days), the DG-blended specimens exhibited lower compressive strength compared to the plain sample. This reduction is attributed to the retardation of early hydration reactions caused by DG. The plain sample showed continuous strength development over time, reaching 26.4 MPa at 28 days. Notably, the specimen with 5% DG (CFDG05) achieved the highest 28-day strength of 29.2 MPa. While the incorporation of 5% DG clearly improves the 28-day compressive strength, the observed reduction in early-age strength and the sharp decline in strength at higher DG levels point to a very narrow optimal range for DG addition. This behavior reflects a delicate balance between the beneficial and detrimental effects of SO_3_ and free CaO introduced through DG. At a moderate level (5%), DG likely provides a controlled supply of sulfate ions, facilitating the formation of stable ettringite and enhancing the densification of the microstructure over time. However, the initial retardation of hydration, evidenced by the lower early-age strength, suggests that DG delays the formation of strength-contributing hydration products in the first week of curing.

However, when the DG content exceeded 5%, a substantial reduction in compressive strength was observed at all curing ages. The 28-day strength of the specimen with 10% DG was slightly reduced to 26.8 MPa, whereas specimens with 15% and 20% DG (CFDG15 and CFDG20) showed significantly lower strength of 14.2 MPa and 12.2 MPa, respectively. More critically, beyond 5% DG, the dramatic reduction in compressive strength across all ages reveals that the system becomes sulfate-rich, which shifts the hydration equilibrium unfavorably. Excessive ettringite formation may not only increase the porosity due to volume expansion but may also block the ongoing hydration of calcium silicate phases, leading to an underdeveloped C–S–H matrix. This is consistent with the significant strength losses observed at 15% and 20% DG, where the 28-day strength was less than half that of the optimum. Furthermore, the narrow window between beneficial and detrimental DG content highlights a critical challenge in practical applications—even small deviations in the DG dosage can result in large variations in mechanical performance.

These findings indicate that achieving a durable and high-strength CFBC ash-based binder using DG requires the precise optimization and stringent quality control of the DG content. Future studies should focus on systematically mapping this narrow optimal range across different curing regimes and mix designs, as well as investigating the synergistic use of DG with other mineral admixtures to broaden the performance window.

### 3.3. Microstructure of CFBC Ash Paste Containing DG

To assess the influence of DG incorporation on the pore structure development of CFBC ash pastes, MIP tests were conducted at curing ages of 7, 14, and 28 days. Two types of measurements were performed: incremental intrusion (mL/g), which reflects the pore size distribution, and cumulative intrusion (mL/g), which represents the total accessible porosity. These were measured for CFBC ash paste specimens with DG content of 0%, 5%, 10%, 15%, and 20%. Figure 12 displays the incremental intrusion profiles over time. In general, all specimens exhibited a reduction in larger pores with increasing curing ages. Specifically, as shown in Figure 12, specimens such as the plain sample, CFDG05, and CFDG10 showed a clear trend in which larger pores decreased while finer pores increased over time. This effect was most pronounced in CFDG05. Specimens with higher DG content, namely CFDG15 and CFDG20, also showed similar trends but to a lesser extent. The overall reduction in pore size with the curing age can be attributed to the continued hydration of the CFBC ash, during which hydration products such as C–S–H and C–A–H are formed and progressively fill the capillary pores, thereby reducing both the porosity and average pore diameter.

Figure 13 presents the cumulative intrusion curves of each specimen over time. As illustrated, all samples showed a decreasing trend in cumulative intrusion with the curing age, regardless of the DG content. This corresponds to a reduction in total porosity and is consistent with the compressive strength results shown in Figure 13, indicating progressive microstructural densification due to the continued hydration of the CFBC ash. For the plain sample, the cumulative intrusion decreased from 0.307 mL/g at 7 days to 0.282 mL/g at 28 days. However, the extent of this reduction varied with the DG content. In specimens with 5–10% DG, the cumulative intrusion values at 28 days were similar to or slightly higher than those of the plain sample, suggesting a marginal increase in total porosity. In contrast, specimens with higher DG content (15–20%) exhibited consistently higher cumulative intrusion values throughout the curing period. Notably, CFDG20 recorded the highest porosity at 28 days with 0.299 mL/g. This trend is attributable to the free CaO present in DG. A moderate amount of free CaO can promote early hydration reactions and enhance the microstructure through Ca(OH)_2_ formation [43]. However, excessive DG incorporation may impair the microstructure and reduce the compressive strength, as shown in Figure 11. This degradation is due to the reduction in CFBC ash content and the excessive presence of free CaO [44]. These findings demonstrate that, while moderate DG addition has a limited influence on the pore structure, excessive DG use significantly increases the total porosity, thereby compromising the mechanical strength and durability of the material.

Based on the results of this study, further research on the long-term durability of CFBC ash-based binders incorporating DG is deemed necessary. In particular, performance verification under harsh environmental conditions—such as sulfate attack, carbonation, and freeze–thaw cycles—should be prioritized. Through such studies, a deeper understanding of the role of DG in regulating the hydration kinetics and enhancing the durability of CFBC ash-based systems can be achieved, which is expected to contribute to the development of high-performance, low-carbon binders for sustainable construction.

## 4. Conclusions

This study experimentally investigated the intrinsic hydraulic properties of CFBC ash and the influence of DG on its hydration behavior. A comprehensive analysis was conducted to evaluate the heat evolution, hydration products, setting behavior, compressive strength development, and microstructural changes of CFBC ash-based binders.

The hydration reaction of CFBC ash exhibited distinct characteristics compared to OPC. Isothermal calorimetry revealed that, unlike OPC, CFBC ash displayed a pronounced first exothermic peak upon contact with water but no distinct second peak. This suggests a deviation from conventional C_3_S-dominated hydration. The magnitude of the first peak slightly increased with higher DG content, likely due to the presence of free CaO in DG. CFBC ash contains inherent free CaO, which reacts with water to form Ca(OH)_2_. This compound subsequently reacts with alumina and silica components to form ettringite and C–S–H gel, contributing to the self-cementing behavior of the material. The XRD analysis at an early age (7 days) showed that increasing DG content resulted in the greater formation of anhydrite and portlandite, attributable to the hydration of DG’s free CaO. At 28 days, anhydrite was observed to transform into dihydrate gypsum via hemihydrate intermediates. The dihydrate phase then reacted with portlandite, Al_2_O_3_, and H_2_O, leading to the increased formation of ettringite. This was confirmed by a higher ettringite peak and reduced anhydrite and portlandite peaks at 28 days. The setting time measurements indicated that both the initial and final setting times increased with greater DG content, confirming that DG has a retarding effect on the setting behavior of CFBC ash paste.

Compressive strength testing revealed that the DG content significantly affected the mechanical properties of the CFBC ash mortar. At early curing ages, the suppressive effect of DG on hydration led to reduced strength compared to the plain sample. At 28 days, however, the sample with 5% DG (CFDG05) achieved the highest strength of 29.2 MPa, suggesting that moderate DG incorporation can effectively regulate the hydration kinetics and enhance the strength. In contrast, DG content above 5% led to marked reductions in compressive strength across all curing ages, indicating that excessive DG negatively impacts binder performance by hindering the formation and densification of hydration products. Mercury intrusion porosimetry (MIP) demonstrated that large pores decreased and fine pores increased with the curing age, particularly in the CFDG05 specimen. This trend is attributed to the gradual filling of pores by hydration products such as C–S–H and C–A–H, resulting in microstructural densification.

Moderate amounts of free CaO promote early hydration and improve the microstructure through Ca(OH)_2_ formation. However, excessive free CaO, resulting from high DG content, leads to increased total porosity and microstructural degradation, which in turn reduce the compressive strength. These findings clearly indicate that, while moderate DG addition can enhance the performance of CFBC ash-based binders, excessive DG incorporation significantly increases the porosity and compromises both the mechanical strength and durability. Therefore, careful optimization of the DG dosage is essential for the effective utilization of CFBC ash in sustainable binder systems. To promote the use of CFBC ash as a low-carbon binder for sustainable construction, future research is essential on blended systems incorporating CFBC ash, as well as conducting field tests and performing durability tests under aggressive environments.

## Figures and Tables

**Figure 1 materials-18-02731-f001:**
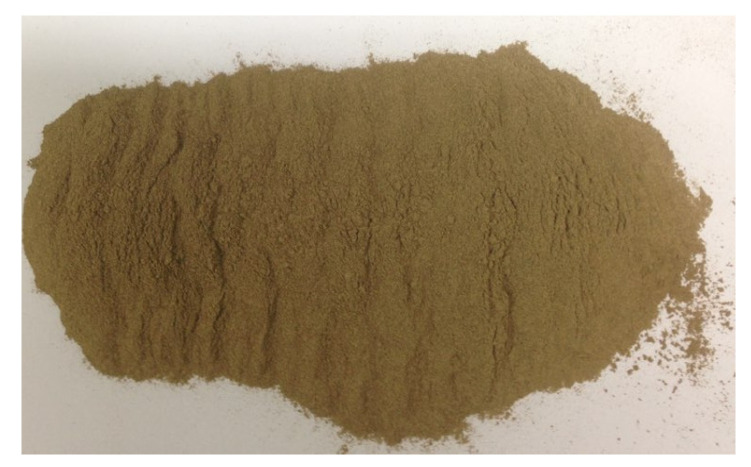
Optical image of CFBC ash.

**Figure 2 materials-18-02731-f002:**
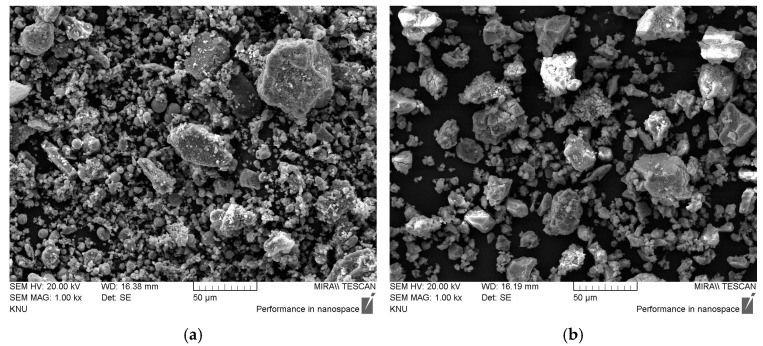
SEM images of raw materials: (**a**) CFBC ash; (**b**) DG.

**Figure 3 materials-18-02731-f003:**
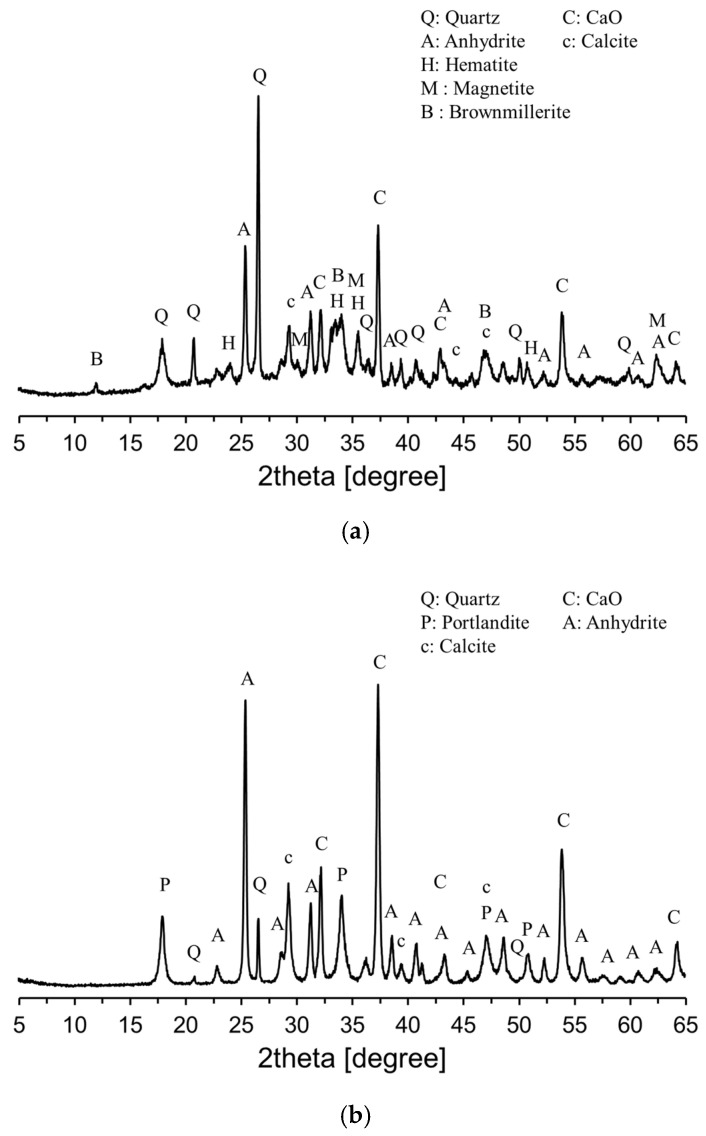
XRD patterns of raw materials: (**a**) CFBC ash; (**b**) DG.

**Figure 4 materials-18-02731-f004:**
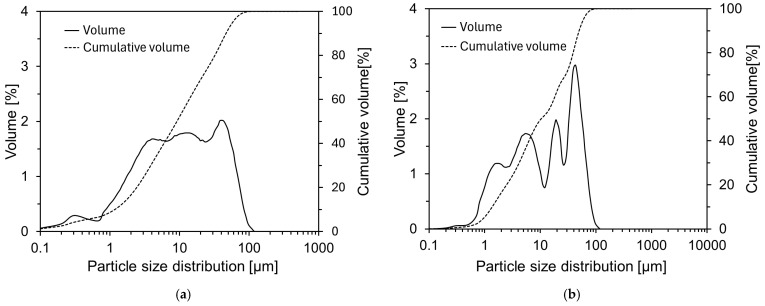
Particle size distributions of raw materials: (**a**) CFBC ash; (**b**) DG.

**Figure 5 materials-18-02731-f005:**
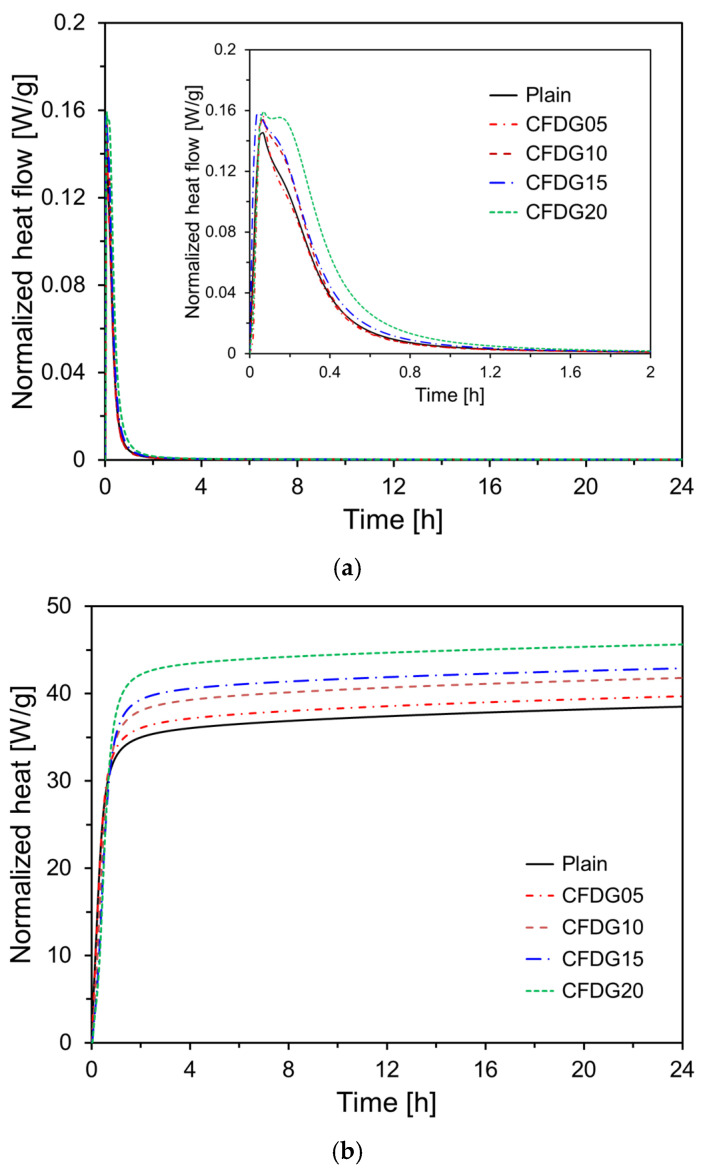
Isothermal calorimetry results: (**a**) normalized heat flow; (**b**) normalized heat.

**Figure 6 materials-18-02731-f006:**
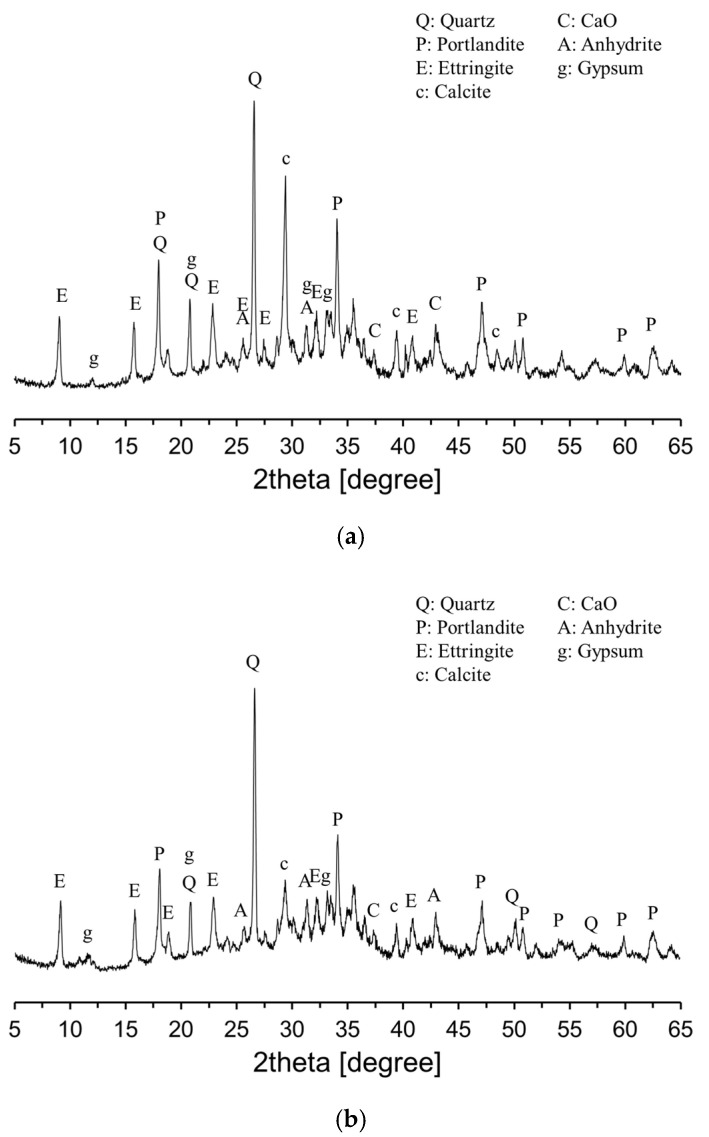
XRD patterns of plain sample: (**a**) 7 days; (**b**) 28 days.

**Figure 7 materials-18-02731-f007:**
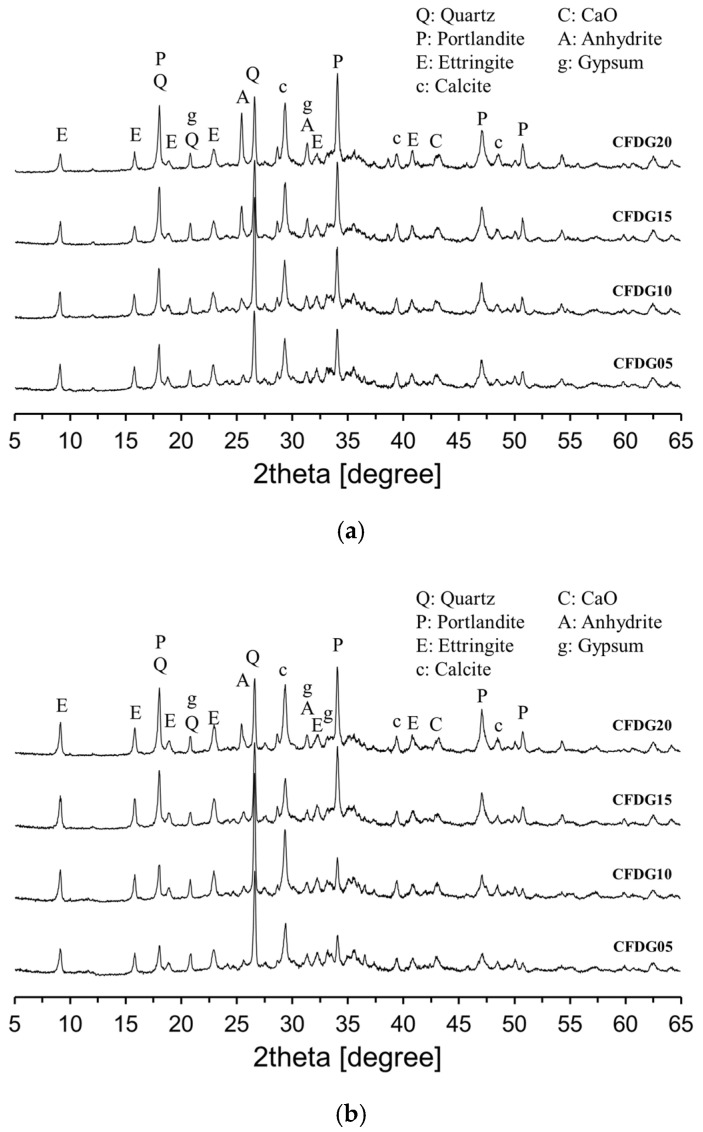
XRD patterns of specimens containing DG: (**a**) 7 days; (**b**) 28 days.

**Figure 8 materials-18-02731-f008:**
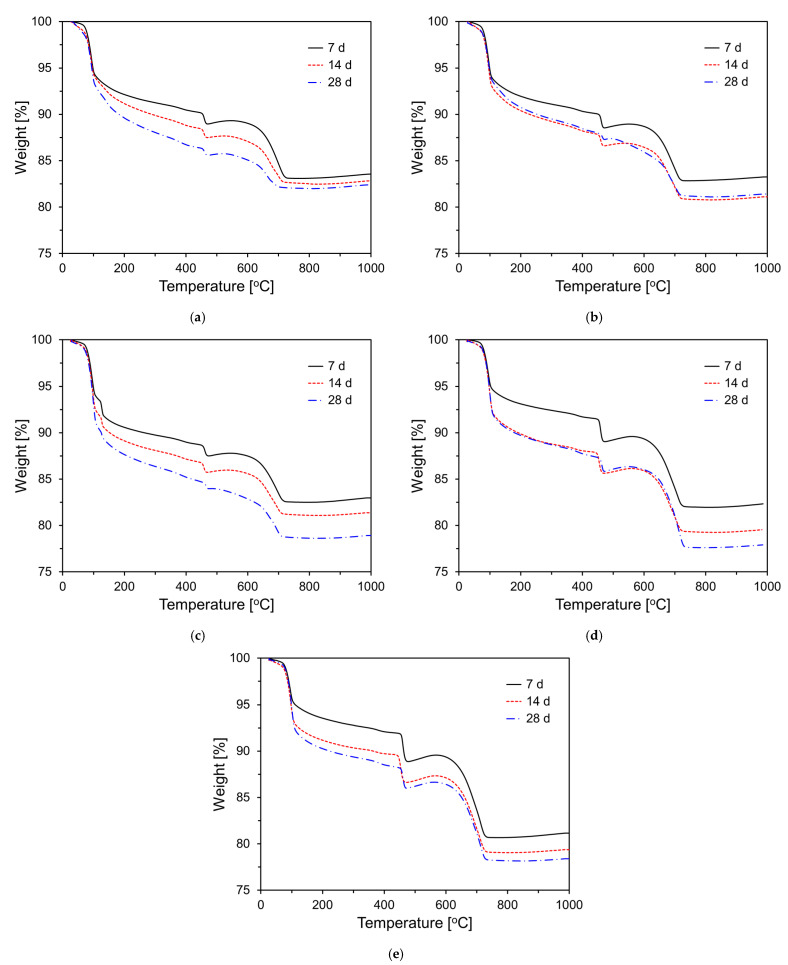
TG curves of specimens with varying DG content at different curing ages: (**a**) plain; (**b**) CFDG05; (**c**) CFDG10; (**d**) CFDG15; (**e**) CFDG20.

**Figure 9 materials-18-02731-f009:**
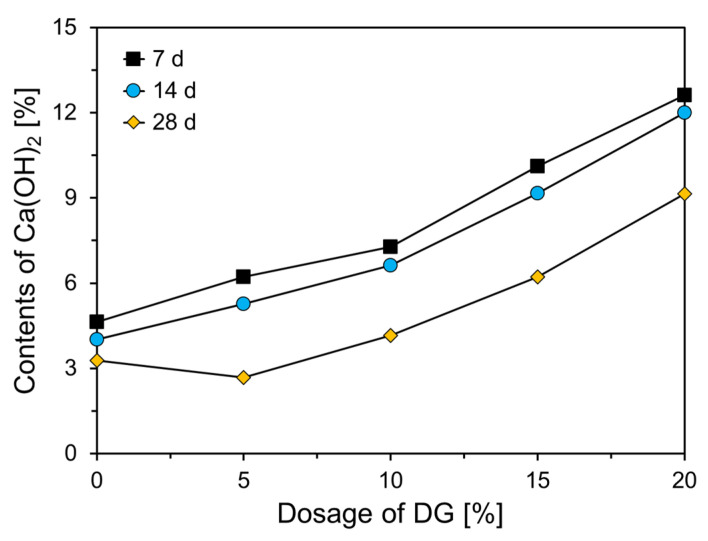
Quantitative analysis of Ca(OH)_2_ content from thermogravimetric data.

**Figure 10 materials-18-02731-f010:**
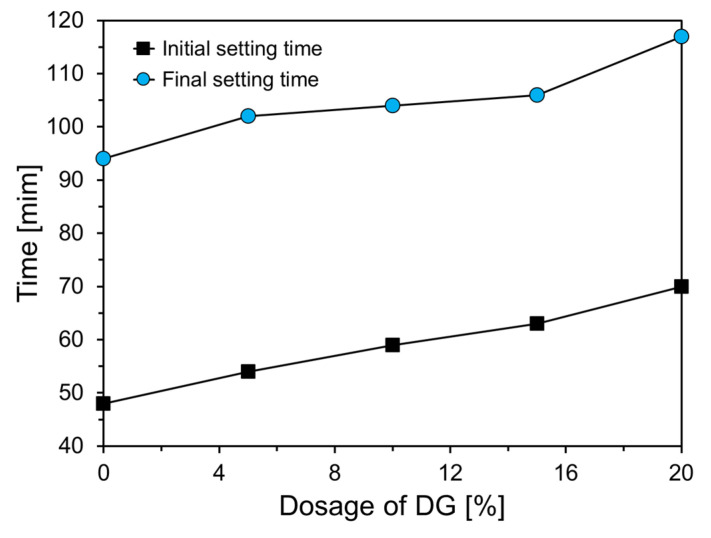
Setting times of CFBC ash pastes with varying DG content.

**Figure 11 materials-18-02731-f011:**
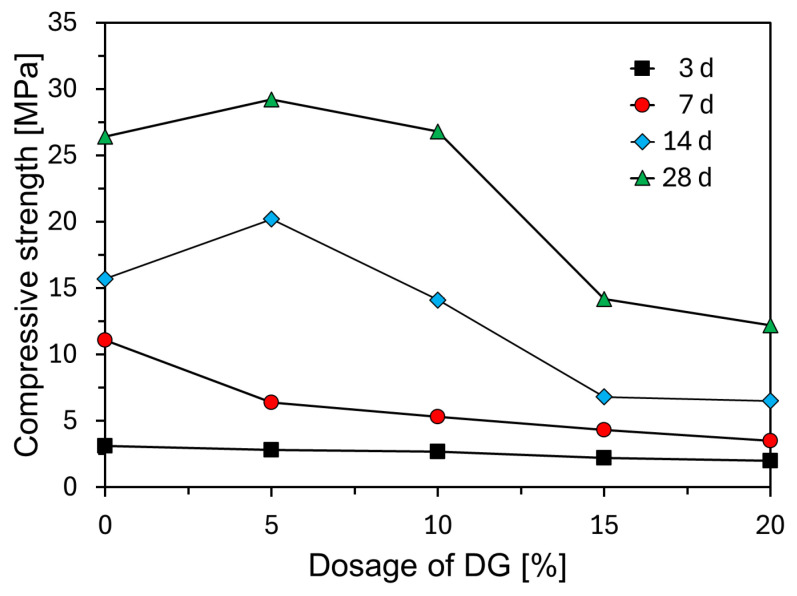
Compressive strength of CFBC ash mortars with varying DG content over curing time.

**Figure 12 materials-18-02731-f012:**
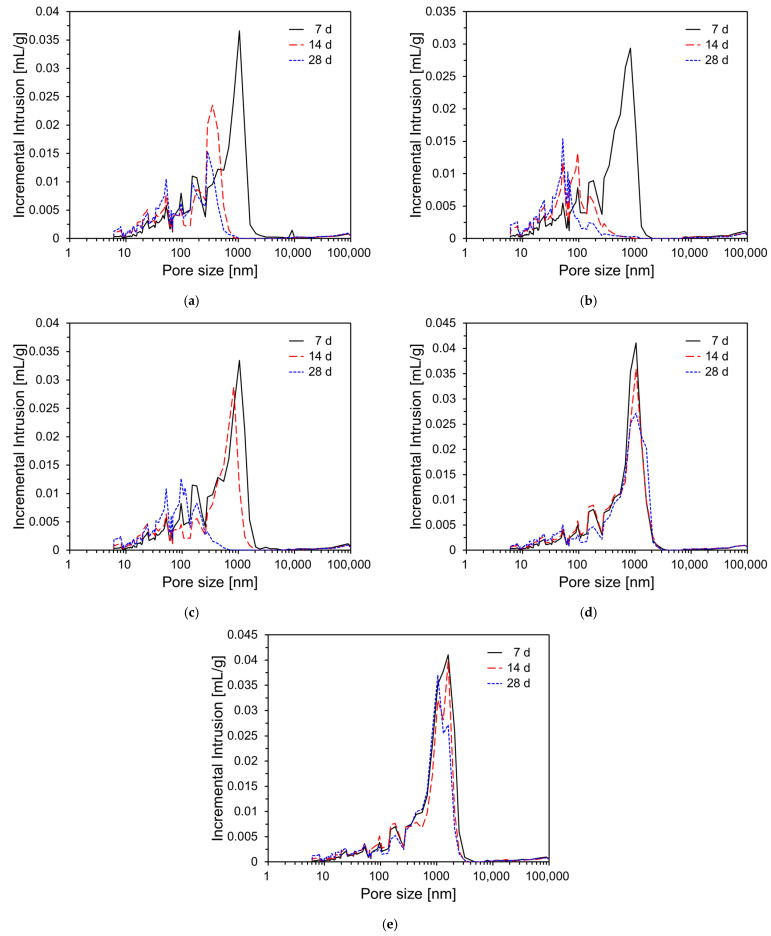
Incremental intrusions of CFBC ash pastes with DG over curing time: (**a**) plain; (**b**) CFDG05; (**c**) CFDG10; (**d**) CFDG15; (**e**) CFDG20.

**Figure 13 materials-18-02731-f013:**
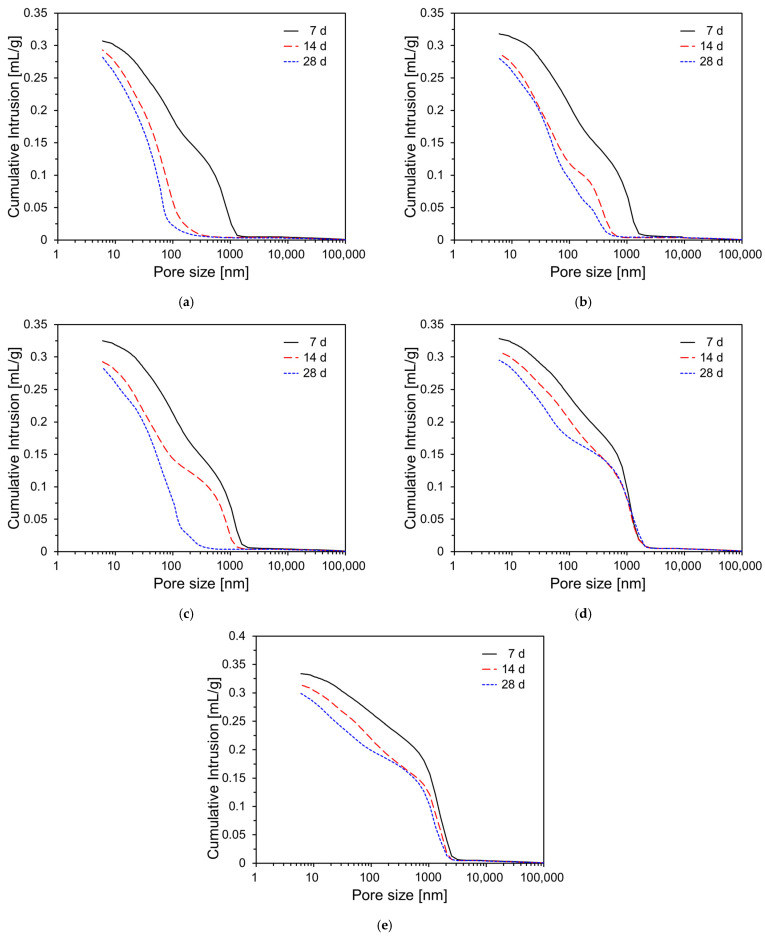
Cumulative intrusions of CFBC ash pastes with DG over curing time: (**a**) plain; (**b**) CFDG05; (**c**) CFDG10; (**d**) CFDG15; (**e**) CFDG20.

**Table 1 materials-18-02731-t001:** Chemical oxide compositions of CFBC ash and DG.

	Chemical Oxide Composition (wt%)								
	SiO_2_	Al_2_O_3_	Fe_2_O_3_	CaO	MgO	K_2_O	Na_2_O	SO_3_	LOI
CFBC ash	27.1	11.3	11.1	35.5	4.6	0.8	1.2	5.6	1.9
DG	4.9	0.4	0.3	62.5	1.7	0.2	-	22.8	7.0

**Table 2 materials-18-02731-t002:** Mixture proportions of specimens containing CFBC ash and DG.

Specimen	W/B (−)	CFBC Ash (g)	DG (g)	Sand (g)
Plain	0.5	100	-	300
CFDG05		95	5	
CFDG10		90	10	
CFDG15		85	15	
CFDG20		80	20	

## Data Availability

The original contributions presented in this study are included in the article. Further inquiries can be directed to the corresponding author.

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
