# Peer review of "Hydration and Mechanical Properties of Low-Carbon Binders Using CFBC Ash"

_materials, 2025, doi:10.3390/ma18122731_

Round 1
Reviewer 1 Report
Comments and Suggestions for Authors
In this manuscript the authors investigated the influence of desulfurization gypsum on the self-cementing behavior of CFBC ash in order to make effective use of CFBC ash in sustainable binder system. The manuscript addresses an important issue. Overall, the technical contents of the paper are of quite high standard. However, the quality of paper should be further enhanced by addressing the following:
(1)Keywords: flue-gas desulfurization gypsum is not mentioned in the passage, so it should be changed to desulfurization gypsum.
(2)Introduction: The example in the sixth line of the third paragraph lacks logic. “Some studies ...Sheng et al. ...” Why is there no specific example of how particle size affects it?
(3)Introduction: There is a lack of foreign data in the background. This part of the logic is a little confused, so I suggest that the author reorganize it and organize the examples, and add some foreign literature data
(4)Figure 2 (b): “ In contrast, the DG particles predominantly appear as platy structures.” Are you sure it is platy structures?
(5)In Figure 7 (a), it can not be seen that the gypsum peak increases with the increase of DG content near 25.5℃. I suggest the author revise the picture.
(6)Figure 8: Figure 8 shows the TG curves of CFBC ash pastes with varying DG content at different curing ages. The title of Figure 8 is not accurate enough and should be revised
(7)The next step of the experiment and the prospect of the results are missing at the end of the article. It is suggested to add them
Reviewer 2 Report
Comments and Suggestions for Authors
The article entitled "Hydration and mechanical properties of low-carbon binders using CFBC ash" is focused on fabrication of series of CFBC-DG samples with DG (desulfurization gypsum) content from 0 to 20% and study the influence of DG content on the self-cementing behavior of CFBC ash. Also, the microstructure of samples and their hydration and mechanical characteristics have been studied. The nature of hydration and self-cementing behavior of CFBC-DG samples have been revealed. The interesting result has been obtained at compressive strength tests of samples. The sample with 5% DG (CFDG05) achieved the highest strength while the excessive DG content in the samples (more than 5%) negatively impacts binder performance. The hypotheses that moderate DG incorporation can effectively regulate hydration kinetics and enhance strength have been formulated. From practical aspect, the suggested CFBC-DG material may be regarded as promising alternative material for cement replacement. Overall, the work is well written, and only minor corrections should be done.
- The Introduction section might be improved by adding some latest (2020-2025) reports emphasizing the role of DG additive on characteristics of low-carbon binders.
- The Materials section should be improved. What is the LOI given in Table 1? Also, the phrase «The high CaO and SO3 contents in CFBC ash (page 3)»seems to be incorrect, because the SO3 content is only 5.57% for CFBC ash. Also, there is a possibility to formation not only CaSO4, but also MgSO4 and Na2SO4 How their existence effect on hydration behavior of blank CFBC.
- The Result and Discussion section might be improved. In order to best assign the role of DG the heat test of blank DG without CFBC ash might be performed. Also, the additional peak at ≈38ͦ (near c) is observed on XRD patterns of samples with DG content more than 10%. Please justify.
Reviewer 3 Report
Comments and Suggestions for Authors
I have reviewed the manuscript titled "Hydration and Mechanical Properties of Low-Carbon Binders Using CFBC Ash." This article is appropriately aligned with the scope of Materials MDPI. Below are several observations that should be considered in the revised version of the manuscript:
- The "Abstract" section lacks a clear presentation of the research hypotheses. It should incorporate more informative points summarizing the research aims, significance, and needs.
- In the introduction, the objective of the study is ambiguous, and the manuscript does not provide a clearly articulated hypothesis, which is a fundamental requirement in scientific research. It is imperative to identify specific international needs to clearly indicate potential beneficiaries.
- In the "Materials and Methods" section, The test methods in Mixture Proportions and Test Methods offer useful insights but have notable limitations. The fixed water-to-binder ratio and limited range of desulfurization gypsum (DG) restrict understanding of optimal mixes. Furthermore, laboratory conditions may not accurately reflect field performance, and the absence of workability tests and the use of small specimens limit the practical relevance of the findings. Solely relying on mercury intrusion porosimetry (MIP) and short calorimetry may not fully capture the microstructural or long-term hydration behavior.
- The "Test Results and Discussion" section presents findings clearly but lacks depth in interpretation. In Section 3.2, while 5% DG improves 28-day strength, the reduction in early-age strength and the sharp decline at higher DG levels indicate a narrow optimal range that deserves more critical analysis. In Section 3.3, the exclusive use of MIP limits the accuracy of pore structure analysis. The slight changes in porosity at low DG levels do not fully explain the strength differences, suggesting that unexamined microstructural factors may be at play. Overall, the discussion is too general and should be expanded to include comparisons with existing studies, explore interdependencies between results, and provide a more comprehensive critical evaluation.
- The "Conclusions" section summarizes the experimental findings effectively, particularly noting the hydration characteristics of CFBC ash and the impact of DG content on mechanical and microstructural properties. Recommendations for future work, such as studying blended systems, conducting field trials, or exploring durability under aggressive environments, are notably absent and should be included to enhance the research's relevance and forward-looking value. The research hypothesis established at the beginning should also be verified in this section.
